

# A lightweight broadband cavity-enhanced spectrometer for NO₂ measurement on uncrewed aerial vehicles

Caroline C. Womack[1,2], Steven S. Brown[2,3], Steven J. Ciciora[2], Ru-Shan Gao[2], Richard J. McLaughlin[2], Michael A. Robinson[1,2], Yinon Rudich[4], and Rebecca A. Washenfelder[2]

[1]Cooperative Institute for Research in Environmental Sciences, University of Colorado, Boulder, CO 80309, USA
[2]Chemical Sciences Laboratory, National Oceanic and Atmospheric Administration, Boulder, CO 80305, USA
[3]Department of Chemistry, University of Colorado, Boulder, CO 80309, USA
[4]Department of Earth and Planetary Sciences, Weizmann Institute of Science, Rehovot 7610001, Israel

*Correspondence to*: Caroline C. Womack (caroline.womack@noaa.gov)

**Abstract.** We describe the design and performance of a lightweight broadband cavity-enhanced spectrometer for measurement of NO₂ on uncrewed aerial vehicles and light aircraft. The instrument uses an LED centered at 457 nm, high-finesse mirrors (reflectivity=0.999963 at 450 nm), and a grating spectrometer to determine optical extinction coefficients between 430–476 nm, which are fit with custom spectral fitting software and published absorption cross sections. The instrument weighs 3.05

kg and has power consumption less than 35 W at 25 °C. A ground calibration unit provides helium and zero air flows to periodically determine the reflectivity of the cavity mirrors using known Rayleigh scattering cross sections. The precision (1σ) for laboratory measurements is 43 ppt NO₂ in 1 s and 7 ppt NO₂ in 30 s. Measurement of air with known NO₂ mixing ratios in the range of 0–70 ppb agreed with the known values within 0.3% (slope=0.997±0.007; r²=0.99983). We demonstrate instrument performance using vertical profiles of NO₂ mixing ratio acquired onboard an uncrewed aerial vehicle between 0–

120 m above ground level in Boulder, Colorado.

## 1 Introduction

The availability of uncrewed autonomous vehicles for land, air, and sea has the potential to improve environmental sampling by allowing better geographical and spatial coverage at lower cost than crewed platforms. Uncrewed aerial vehicles (UAVs) can be divided into five categories based on their weight, including nano (<0.250 kg), micro (0.25–2 kg), small (2–25 kg) and

medium (25–150 kg), and large (>150 kg). Even the largest UAVs have limited payloads compared to crewed aircraft, and require lightweight instruments with low power consumption.

Miniaturized research-grade atmospheric instruments that weigh less than ~5 kg have important potential for deployment on small and medium UAV platforms. Lightweight sampling payloads have already been demonstrated for UAVs (Ramana et al.,

2007; Telg et al., 2017). Existing miniaturized aerosol instruments include a condensation particle counter for aerosol size distribution (Model 9403; Brechtel Manufacturing Inc., Hayward, CA, USA) (Bates et al., 2013), an optical particle counter



for aerosol size distribution (Gao et al., 2016), a sun photometer for solar irradiance and sky radiance (Murphy et al., 2016), and a three-wavelength absorption photometer for light absorption coefficients (Model 9406; Brechtel Manufacturing Inc., Hayward, CA, USA) (Bates et al., 2013). Many miniaturized gas-phase instruments exist, including for methane (e.g., Nathan

et al., 2015), $CO_2$ (Zhao et al., 2022), and ozone (e.g., Deshler et al., 2008; Kezoudi et al., 2021) with varying accuracy and detection limits depending on the detection technique.

Among the potential target gas species, accurate measurements of nitrogen oxide (NO) and nitrogen dioxide ($NO_2$) concentrations are crucial due to their role in atmospheric photochemical oxidation. Most $NO_2$ in the lower troposphere is

oxidized from NO emitted from fossil fuel combustion and biomass burning, and $NO_2$ and NO typically photochemically equilibrate within a few minutes (Masson-Delmotte et al., 2021). Smaller sources of tropospheric $NO_x$ (NO + $NO_2$) include soils and lightning (Masson-Delmotte et al., 2021). Characterizing horizontal and vertical $NO_2$ concentration gradients is important due to its heterogeneous sources and variable lifetime. Additionally, there is a need for *in situ* $NO_2$ measurements to validate remote sensing methods, particularly those available from recent and planned satellite instruments, such as the

TROPOspheric Monitoring Instrument (TROPOMI), the Geostationary Environmental Monitoring Spectrometer (GEMS), and Tropospheric Emissions Monitoring of Pollution (TEMPO).

$NO_2$ instruments with parts-per-trillion (ppt) precision, accuracy of a few percent, linear response over two to three orders of magnitude, and ~1 s time response are needed for satellite validation, air quality monitoring, and atmospheric studies.

Successful field instruments that meet these criteria use laser-induced fluorescence (e.g., Thornton et al., 2000); cavity-attenuated phase shift spectroscopy (e.g., Kebabian et al., 2005); cavity ring-down spectroscopy (e.g., Wild et al., 2014); broadband cavity-enhanced spectroscopy (e.g., Min et al., 2016); or conversion to NO with subsequent detection by chemiluminescence (e.g., Ryerson et al., 2000) or laser-induced fluorescence. However, the current implementations of these instruments are too large and heavy to be deployed onboard UAVs, and some have power consumption requirements that

exceed what can be supplied by batteries. Small, lightweight electrochemical $NO_2$ sensors exist, but lack the desired precision, time response, and accuracy for scientific field studies and can be affected by chemical interferences, relative humidity, and temperature (Williams et al., 2014). Broadband cavity enhanced spectroscopy and cavity ring-down spectroscopy have great potential for miniaturization due to their relative simplicity, small set of required components, and modest power and pump requirements. For example, a commercial cavity ring-down instrument for aerosol extinction has been developed recently that

weighs 7.7 kg and measures 0.5 m × 0.3 m × 0.2 m (Optical Extinction Analyzer; Nikira Labs, Mountain View, CA, USA).

In this work, we describe the Miniature Airborne broadband Cavity-Enhanced Spectrometer (mACES), that weighs 3.05 kg and measures $NO_2$ with a precision (1σ) of 43 ppt in 1 s. The instrument design and reduced weight allow it to be operated onboard a small rotary-wing UAV to measure spatial distributions of $NO_2$ in the lowest part of the troposphere. We present

the precision and accuracy of the $NO_2$ instrument, along with measurements of ambient $NO_2$ acquired between 0–120 m above



ground level during test flights onboard a DJI Matrice 600 Pro UAV. Finally, we discuss the use of this instrument in future field deployments and potential improvements to further reduce the instrument weight and improve the measurement precision.

## 2 Instrument design

Broadband cavity enhanced spectrometers (BBCES) are used at visible and ultraviolet wavelengths to measure aerosol extinction or structured absorption by gases (Fiedler et al., 2003; Washenfelder et al., 2008). BBCES instruments consist of a broadband light source coupled to an optical cavity with the output measured by a grating spectrometer. Field measurements of $NO_2$ from ground and aircraft using BBCES have been described previously (Kennedy et al., 2011; Washenfelder et al., 2011b; Min et al., 2016; Zarzana et al., 2017). We designed a miniaturized version of our aircraft BBCES instrument (Min et al., 2016) with reduced size and weight for operation on a UAV platform while maintaining measurement precision and accuracy. The weight of the instrument has been reduced to 3.05 kg with a power consumption of 15–35 W, depending on the ambient temperature and the cooling requirements for the light source and spectrometer. A summary of the instrument specifications is given in Table 1. The optical system, flow system, mechanical mounting, and data acquisition are described in detail below.

## 2.1 Optical system

The optical system is shown in Fig. 1a, and consists of a light-emitting diode (LED), off-axis parabolic mirror, optical cavity, bandpass filter, collection lens, optical fiber, and grating spectrometer with charge-coupled device (CCD) array detector. An LED centered at 457 nm with full-width at half-maximum of 15 nm (LZ1-00B202; LEDEngin Inc., San Jose, CA, USA) is powered by a custom constant-current power supply (3.7 VDC at 1.0 A) and temperature-controlled at 22.5±0.05 °C using a thermoelectric cooler (TEC; CP60233, CUI Devices, Tualatin, OR, USA). The LED light is collected with an off-axis parabolic mirror (50328AL, 2.0 cm effective focal length; Newport Corp., Irvine, CA, USA) and free-space coupled into a 22.3 cm long cavity formed by two 2.5 cm diameter, 0.5 m radius of curvature mirrors (FiveNines Optics, Boulder, CO, USA), with measured reflectivity of 0.999963 at 450 nm. Light output from the cavity is filtered with a bandpass filter (FF01-452/45-25; Semrock Inc., Rochester, NY, USA) and coupled into a 1 m long circular optical fiber (600 μm diameter; Ocean Insight Inc., Dunedin, FL, USA). Stray light is minimized using the bandpass filter and baffling. The optical system is initially rough-aligned using a small HeNe laser mounted on the laboratory bench. The mirror mounting plates are then finely aligned on their carbon support rods using fixed clamps with set screws to maximize cavity throughput before locking the mounting plates in their final position.

The spectrum is measured by a grating spectrometer (QE Pro; Ocean Insight Inc., Dunedin, FL, USA) with a 200 μm wide entrance slit and a 1024 × 58 array of 18-bit pixels. The spectral region spans 384.3–499.9 nm with an average full-width at half-maximum (FWHM) resolution of 0.9 nm across the entire spectral region. The integration time for each spectrum is 0.15





s. The QE Pro is more limited in its spectral lineshape and dark noise than the SP2150 spectrometer and PIXIS2KBUV CCD (Princeton Instruments, Trenton, NJ, USA) used in our aircraft instrument (Min et al., 2016). However, the QE Pro weighs only 1.15 kg compared to 6.8 kg, requires no physical shutter, and can read out the vertically-integrated CCD in 0.002 s.

## 2.2 Flow system

The flow system is shown in Fig. 1b and consists of a filter, optical cavity, pressure sensor, flow sensor, and pump. Aerosol particles are removed by a single-stage filter assembly (401-21-25-50-21-2; Savillex, Eden Prairie, MN, USA) with replaceable 0.45 μm pore polytetrafluoroethylene (PTFE) filters (450-25-2; Savillex, Eden Prairie, MN, USA). During the test flights described here, the sampling inlet was a 0.635 cm OD Teflon tube that extended directly above the UAV. The optical cavity is constructed from PTFE (1.90 cm ID) and the sample flow enters and exits through PTFE Teflon fittings. $NO_2$ has negligible losses on Teflon (Fuchs et al., 2009; Min et al., 2016). The cavity mirrors are sealed using o-rings on their face and a mirror holder that compresses them against the mounting plate. Following the approach described in Min et al. (2016), mirror purges for the cavity mirrors are not used. Mirror cleanliness is monitored using the measured mirror reflectivity (see Sect. 3), and mirrors are removed and cleaned as necessary, approximately monthly. Sample pressure is measured by a miniature pressure sensor (24PCCFA6A; Honeywell, Golden Valley, MN, USA). Sample flow is measured by a miniature flow sensor (D6F; Omron, Kyoto, Japan) that was calibrated from 0–2.0 volumetric liters per minute (vlpm) (DryCal; Mesa Laboratories Inc., Lakewood, CO 80228). The flow is pulled through the cavity using a small rotary vane pump (G 6/01-K-LCL; Gardner Denver Thomas GmbH, Fürstenfeldbruck, Germany) with a typical flow rate of 1.4 vlpm at 840 hPa. The volume of the cavity is 63 $cm^3$, resulting in a residence time of 2.5 s. Density inside the cavity is determined from the measured pressure and the ambient temperature measured by a thermistor mounted to the outside of the cavity tubing.

## 2.3 Electrical system

The maximum total power consumption of the instrument is 35 W, but varies with ambient temperature because the LED and CCD are temperature-controlled. The instrument is powered by a 14.8 V, 2200 mAh rechargeable Li Ion battery pack (31021; Tenergy Corp., Fremont, CA, USA). The major contributors to the power consumption are the LED, TEC for the LED, and spectrometer. The measured battery lifetime of the full system in the laboratory is 2 h 20 min but is shorter at higher operating temperatures.

## 2.4 Data acquisition hardware and software

The data acquisition system consists of a custom printed circuit board (PCB) that distributes power and acquires signals, and a lightweight, single-board computer with Linux operating system (BeagleBone Black Rev C; BeagleBoard.org, Oakland, MI, USA). The PCB acquires analog and digital inputs from two temperature sensors (measuring the cavity and ambient temperatures), pressure sensor, and flow sensor. The TEC is controlled by a PCB-mountable Peltier Controller module (TEC-1092, Meerstetter Engineering GmbH, Rubigen, Switzerland) and interfaces with the BeagleBone computer via serial

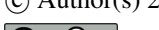



connection to record LED temperature and TEC current. The data acquisition software is written in C/C++, and used the Ocean Insight SeaBreeze API v3.0.11 device driver for embedded platforms to interface with the spectrometer. LED temperature, ambient temperature, sample pressure, sample flow, and spectra are acquired at an integration time of 0.15 s and later averaged to 1 s. However, the sample residence time of 2.5 s in the optical cavity is the limiting factor for the time resolution. On the ground, communication with the BeagleBone computer uses an HTML interface to the Linux operating system.

## 2.5 Mechanical system

The mechanical assembly of the instrument is shown in Fig. 1c. The optical, flow, and electrical components are mounted to a cage system consisting of 1.27 cm diameter hollow carbon fiber rods (GR-CFR-TUBE-0.500OD; GraphiteStore, Northbrook, IL, USA). Optical components are attached to the rods using custom-designed aluminum plates (0.76 cm thick). The instrument performance depends on the stability of the optical alignment, and all mechanically-adjustable components were secured by screws or other locking components. The BeagleBone computer, custom electronics board, spectrometer, and batteries are attached to a 0.16 cm thick aluminum sheet. The instrument is attached to the tubular expansion mounting kit of the Matrice 600 Pro UAV using four quick release brackets that are secured with thumb screws. No additional weather-proofing is included in the instrument because the Matrice 600 Pro UAV is not designed to fly in rain, snow, fog, or wind speeds exceeding 8 m s$^{-1}$. The total hardware cost of the mACES instrument is approximately \$20,000, and is dominated by the cost of the spectrometer and the high-finesse cavity mirrors.

## 2.6 Matrice 600 Pro UAV

The Matrice 600 Pro is a small six-rotor UAV (DJI; Shenzhen, China) that weighs 10.0 kg without payload and has a maximum takeoff weight of 15.5 kg. The Matrice 600 Pro is powered by six lithium-ion (22.8 V, 5700 mAh) batteries that support a flight time equal to 38 min minus (3.6 min kg$^{-1}$ × payload mass), which is equivalent to 27 min for the current instrument weight of 3.05 kg. The UAV is specified for operation at temperatures of -10–40 °C, wind speeds less than 8 m s$^{-1}$, and dry conditions. Its maximum altitude is 2500 m above ground level, with maximum ascent and descent velocities of 5 m s$^{-1}$ and 3 m s$^{-1}$ respectively, although current US FAA regulations restrict UAV flight in Class G airspace to 120 m above ground unless a waiver is obtained. With the propellers and frame arms unfolded, it measures 1.7 m × 1.5 m × 0.7 m. An onboard computer records position, altitude, and auxiliary data. The Matrice 600 Pro is actively controlled by an operator with line-of-sight communication, although the onboard avionics allow for stability and landing.

## 3 Data analysis

The light extinction in the cavity, $\alpha_{ext}(\lambda)$, is calculated following the approach described in Min et al. (2016):



$$\alpha_{ext}(\lambda) = \left( \frac{(1-R(\lambda))}{d} + \alpha_{Ray,ZA}(\lambda) \right) \left( \frac{I_{ZA}(\lambda) - I_{sample}(\lambda)}{I_{sample}(\lambda)} \right) + \Delta\alpha_{Ray}(\lambda) \tag{1}$$

where $\lambda$ is the wavelength of light, $d$ is the cavity length, $R(\lambda)$ is the mirror reflectivity, $\Delta\alpha_{Ray,ZA}(\lambda)$ is the Rayleigh scattering of zero air, $I_{ZA}(\lambda)$ is the reference spectrum of zero air, and $I_{sample}(\lambda)$ is the measured spectrum of ambient air. The term $\Delta\alpha_{Ray}(\lambda)$ is equal to $\Delta\alpha_{Ray,ZA}(\lambda)$ - $\Delta\alpha_{Ray,sample}(\lambda)$, and is needed to explicitly account for pressure differences between the Rayleigh scattering of the reference zero air spectrum, $I_{ZA}(\lambda)$, acquired on the ground and the sample spectrum, $I_{sample}(\lambda)$, acquired on the UAV.

The mirror reflectivity, $R(\lambda)$, in Eqn. 1 can be determined using standard additions of known extinction. In this case, we use the known Rayleigh scattering of helium and zero air. These are added sequentially while the instrument is on the ground, using compressed helium and zero air with a mass flow controller (MC-5SLPM-D-DB15; Alicat Scientific Inc., Tucson, AZ, USA) to overflow the inlet. We use the Rayleigh scattering cross-sections described in Min et al. (2016), which are based on work by Bodhaine et al. (1999), Shardanand and Rao (1977), and Sneep and Ubachs (2005).

The measured extinction, $a_{ext}(\lambda)$, is equal to the sum of the contributing extinctions:

$$\alpha_{ext}(\lambda) = \sum_{i}^{n} \sigma_i(\lambda) N_i + p(\lambda) \tag{2}$$

where $\sigma_i(\lambda)$ and $N_i$ are the absorption cross section and number density of the $i$th gas-phase absorber and $p(\lambda)$ is a 4th-order polynomial that encompasses the broad features in the measured extinction that can be attributed to drifts in the light source intensity, pressure, and spectrometer optics. Values of $\sigma_i(\lambda)$ were taken from high-resolution reference cross sections for CHOCHO (Volkamer et al., 2005), $H_2O$ (Harder and Brault, 1997), and $O_4$ (Greenblatt et al., 1990) and convolved to the measured spectrometer resolution. To improve the quality of the spectral fit, a reference spectrum for $\sigma_{NO2}(\lambda)$ was regularly determined by overflowing a small amount of $NO_2$ from a cylinder (27.2 ppm, diluted in zero air to approximately 100 ppb; Linde Gas & Equipment Inc., Bethlehem, PA, USA) into the cavity. This reference spectrum was scaled to the literature reference spectrum of Vandaele et al. (1998) and then used in the spectral fitting. This minimized the residual features in the fit, and is similar to the process described in Liang et al. (2019). Additionally, this $NO_2$ reference spectrum was used to adjust the spectrometer wavelength calibration, and to determine the spectral lineshape that was convolved with the other literature reference spectra.

The spectral fitting to determine $N_i$ and $p(\lambda)$ in Eqn. 2 used custom software developed in Igor Pro (WaveMetrics Inc., Portland, OR, USA) and based on Levenburg-Marquardt least-squares linear fitting. The fit was optimized between 430 and 476.5 nm and used a wavelength-dependent weighting factor to prioritize the fit in the spectral region where the mirror reflectivity was highest.





## 4 Instrument operation during UAV flights

We developed a standard vertical profile sampling sequence during test flights of the mACES instrument onboard the Matrice 600 Pro UAV. First, we physically attach the mACES instrument to the Matrice 600 Pro expansion mounting kit using the four quick release brackets. We then power on the instrument and record dark background spectra with the LED off. Using the ground calibration unit shown in Fig. 1b, we sequentially overflow the instrument inlet with 2.0 vlpm of helium and zero air for 15 s each. We then overflow the inlet with ~100 ppb of $NO_2$ in zero air to provide the $NO_2$ reference spectra for Eqn. 2.

Finally, we disconnect the ground calibration unit for flight.

The sampling pattern consisted of vertical profiles ascending from 0–120 m, with 10 s hovering at constant altitude at 10 m intervals. The vertical descent was continuous at 0.5 m s$^{-1}$. This sequence requires approximately 7 min, so a single UAV flight can include three vertical profiles for a total flight time of 21 min with a 25% battery power margin for the UAV. Other flight

patterns, such as horizontal sampling, targeted sampling near point sources, or shorter or longer flights are also possible, but were not tested here.

Following the flight sequence, the instrument is again connected to the ground calibration unit to repeat the helium and zero air measurements. Data can be transferred from the BeagleBone computer for offline spectral fitting and data analysis. If

multiple UAV flights are planned, the UAV batteries are replaced, which requires less than 5 min. Following this sequence, we would complete two flights with three vertical profiles each, for a total of six profiles per hour.

## 5 Results and discussion

### 5.1 $NO_2$ measurement accuracy calculated from propagated uncertainties

The instrument accuracy can be evaluated by propagating the uncertainties in Eqn. 1. These include the uncertainty in the

Rayleigh scattering cross-section of zero air (±2%), pressure (±0.1%), temperature (±0.07%), and absorption cross-section of $NO_2$ (±4%). The contribution of the Rayleigh scattering cross-section of He is negligible. Summing these errors in quadrature gives a total calculated uncertainty of ±4.5% for $NO_2$. This does not account for uncertainty imparted by the spectral fitting procedure or the scaling of the measured reference $NO_2$ cross section, as described in Section 3.

### 5.2 $NO_2$ measurement accuracy evaluated with standard additions

The instrument accuracy was also evaluated by comparison to standard additions of $NO_2$. In the laboratory, $O_3$ concentrations were generated and measured by a commercial $O_3$ monitor (49i; ThermoFisher Scientific, Waltham, MA, USA) and subsequently reacted with an excess of 3 ppm NO to quantitatively convert $O_3$ to $NO_2$ (Washenfelder et al., 2011a) which was measured by the mACES instrument. Fig. 2 shows a correlation plot for $NO_2$ concentrations ranging from 0–70 ppb acquired





for 1 min each. The slope is 0.997±0.007 and the intercept is 0.237±0.253 ppt. The $r^2$ value is 0.99983, indicating excellent
agreement between the $NO_2$ standard additions and mACES measurements.

### 5.3 $NO_2$ measurement precision

The instrumental precision was evaluated by measuring zero air in the laboratory over 2 h, with measurements of mirror
reflectivity, spectrometer dark counts, and the $NO_2$ reference spectrum at the start of the measurement period. Allan deviation
plots (Werle et al., 1993) were calculated for both the optical extinction and retrieved $NO_2$ concentrations, to quantify the
precision and drift as a function of time. Fig. 3a and 3b show the Allan deviation for the optical extinction and retrieved $NO_2$
concentrations. The calculated precision (1σ) of the retrieved $NO_2$ is 43 ppt at 1 s and 7 ppt at 30 s. Both the accuracy and
precision are sufficient for most tropospheric measurements of $NO_2$, including measurements of small spatial gradients and
measurements in clean, remote locations.

### 5.4 Vertical profiles acquired onboard a UAV in Boulder, Colorado

Fig. 4a shows the vertical profile of $NO_2$ from 0–120 m above ground level measured by the mACES instrument near the
NOAA David Skaggs Research Center in Boulder, Colorado (39.9905 deg N, 105.2629 deg W) between 12:00 pm - 12:30 pm
local time (MDT) on 26 May 2022. The operational sequence described in Sect. 4 was followed for the test flights. The 0.15 s
spectral data has been averaged to 1 s prior to calculating the light extinction from Eqn. 1 and the average and standard
deviation for each 10 s period at constant altitude are also shown. The corresponding temperature profiles are shown in Fig.
4b.

Detailed computational fluid dynamics (CFD) simulations have been completed for the DJI Matrice 600 Pro UAV and a
sampler mounted below (McKinney et al., 2019). McKinney et al. determined that the propellers draw laminar flow from
above the UAV, which is turbulently recirculated at the propellers and then ejected below the UAV. The authors estimate that
the vertical mixing volume extends 7 m above the UAV, with a vertical bias of approximately -3 m between the physical
position of the sampler and the measured air.

Similar to the McKinney et al. (2019) study, the mACES instrument was mounted below the UAV. The total weight of our
system was 12.65 kg (9.6 kg UAV and 3.05 kg payload) compared to 10.5 kg in McKinney et al (9.6 kg UAV and 0.9 kg
payload), which would require increased propeller speeds and may increase the vertical mixing volume. The mACES inlet was
located vertically above the UAV, in the region that is modeled to have approximately laminar flow. We estimate that each 1
s measurements of $NO_2$ in Fig. 4 represents a vertically-mixed sample that extends approximately 7 m above the UAV.

The vertical $NO_2$ measurements indicate that the boundary layer height exceeded 120 m with well-mixed $NO_2$ concentrations,
as expected for mid-day measurements acquired away from local point sources. Measured $NO_2$ concentrations varied between





0.4–0.6 ppb. The measurement precision of 43 ppt $NO_2$ in 1 s suggests that the observed variability within the vertical profile represents real $NO_2$ variation, which is expected since the measurement site is 440 m from a busy road.

The reflectivity measurements at the beginning and end of the test flight were 0.999954 and 0.999953 at 450 nm, and the signal
intensity on the spectrometer during zero air spectra changed by less than 1% before and after the flight, indicating that the optical alignment of the mACES instrument was stable and unaffected by the vibration of the UAV during the flight.

## 6 Summary and future work

We have demonstrated a miniaturized BBCES instrument that measures $NO_2$ onboard a small UAV. Laboratory measurements of standard $NO_2$ concentrations by this instrument showed a high correlation ($r^2$=0.99983) and accuracy of 0.3%, well within
the 4.5% calculated by propagating component uncertainties. The precision (1σ) during laboratory measurements of zero air was 43 ppt $NO_2$ in 1 s. Measurements of mirror reflectivity and signal intensity before and after a UAV test flight from 0–120 m indicated that the optical system was not affected by physical vibrations. Future improvements in the precision and detection limit could be achieved with brighter LEDs, higher reflectivity cavity mirrors, or improved LED temperature control.

Reducing the instrument weight would allow longer flight durations, and the possibility to move to lighter and cheaper UAV vehicles. Weight reductions in the onboard power system and electronics are possible, including eliminating a DC-DC power converter. Similarly, weight reductions in the flow system are possible, including the possibility of a custom particle filter assembly and smaller diameter teflon tubing and fittings. The weight of the optical cage system could potentially be reduced with smaller diameter carbon fiber rods that allow reduced cage system dimensions or mounting plates constructed from carbon
fiber or other materials that are less dense than aluminum. A custom-built spectrometer could also significantly reduce weight.

Some simple modifications could improve the robustness of the design. A custom, cladded fiber bundle coupled to the spectrometer would protect the optical fiber during flights and potentially improve the light collection. A weather-proof cover would allow flights on other UAVs that can operate in rain, snow, or mist.

An instrument with this accuracy, precision, size, and weight has potential for measuring onboard small UAVs, as well as balloon sondes and being deployed as a distributed network for low-cost monitoring. This $NO_2$ instrument could be deployed together with a selected set of miniature gas, aerosol, or meteorological sensors, such as those described in Telg et al. (2017), for vertical sampling and atmospheric characterization. Further, changing the spectral region of the instrument by changing
the LED, high-finesse cavity mirrors, bandpass filter, and spectrometer grating would allow different target analytes to be measured. These include nitrous acid, formaldehyde, sulfur dioxide, aerosol extinction, and other species that have previously been measured by BBCES.



**Code and data availability**

Code and data used are available upon request (caroline.womack@noaa.gov)

**Author contribution**

CCW, SSB, RSG, MAR, YR, and RAW conceptualized the instrument. RJM and MAR designed and fabricated the mechanical components. SJC designed and fabricated the electrical components. CCW and RAW assembled and tested the instrument, processed the data, and wrote the manuscript. All authors contributed to editing the manuscript.

**Competing interests**

The authors declare that they have no conflict of interest.

**Acknowledgements**

We thank Isaac Vimont and Chelsea Stockwell for their assistance with the UAV and its operation. We thank Laurel Watts, Dori Nissenbaum, and Ofir Shoshanim for helpful discussions. We thank Erin Bobby, Star Fassler, Jake Leicht, Lucas McMahan, Aquzana Mejia and the University of Colorado Mechanical Engineering Senior Design program. CCW and MAR
were partially supported by the NOAA Cooperative Agreement with CIRES, NA17OAR4320101. YR and SBB acknowledge support by the USA-Israel binational Science Foundation (BSF Grant #2020055).




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



**Table 1. Specifications of the lightweight mACES instrument.**

| | |
|---|---|
| LED center wavelength | 457 nm |
| LED spectral width (FWHM) | 15 nm |
| LED optical power output | 1350 mW at 1.0 A |
| Maximum mirror reflectivity | 0.999963 at 450 nm |
| Spectrometer range | 384.2–499.9 nm |
| Spectrometer resolution (FWHM) | 0.9 nm at 455 nm |
| Spectrometer integration time | 0.15 s |
| Sample flow rate | 1.4 vlpm at 840 mb |
| Power consumption | Less than 35 W at 25 deg C |
| Instrument weight with battery | 3.05 kg |
| Method to calibrate cavity loss | Rayleigh scattering of helium and zero air |
| Time resolution | 1 s (2.5 s residence time) |
| Accuracy | ±4.5% |
| Precision | 43 ppt $NO_2$ in 1 s; 7 ppt $NO_2$ in 30 s |






**Figure 1.** (a) Schematic of the optical system, showing the LED, off-axis parabolic mirror, optical cavity, and spectrometer; (b) Block diagram of the flow system, showing the ground calibration unit, inlet filter, sample cell, flow sensor, pressure sensor, and pump; (c) Model of the instrument; (d) Photograph of the instrument.

### (a) Optical system

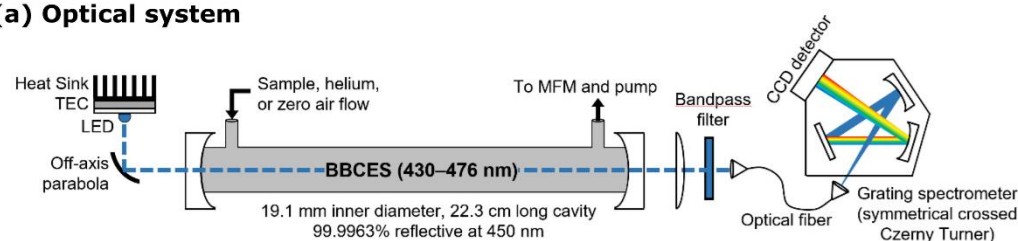

### (b) Flow system

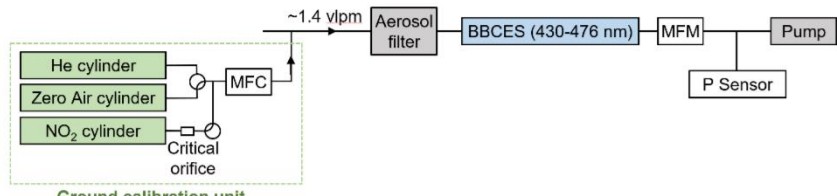

### (c) Model

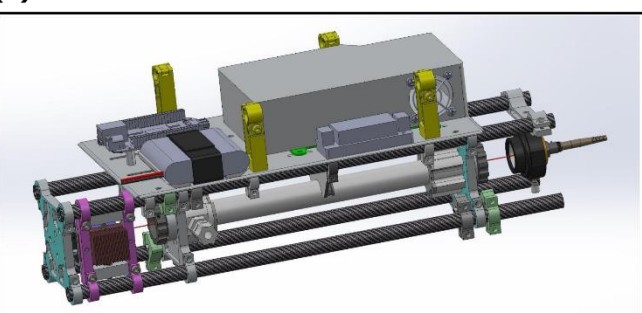

### (d) Photograph

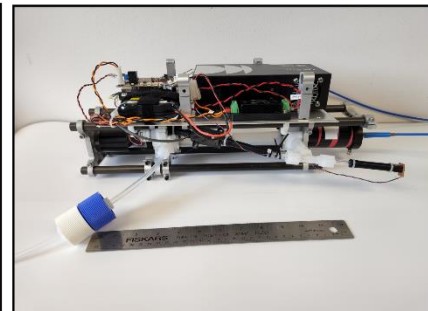





**Figure 2.** Correlation plot showing mACES measurements of $NO_2$ standards generated by quantitative reaction of known amounts of $O_3$ with NO. The grey dashed line shows the 1:1 line.

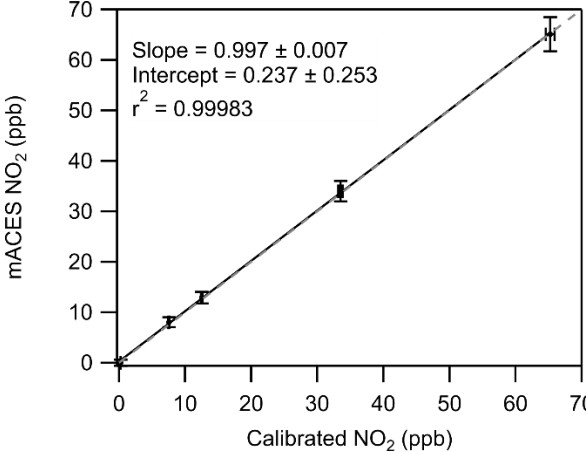





**Figure 3.** (a) Allan deviation plot of fitted $NO_2$ during zero air addition to the cavity (b) Normalized frequency distribution of fitted $NO_2$ during the same zero air measurement time period.

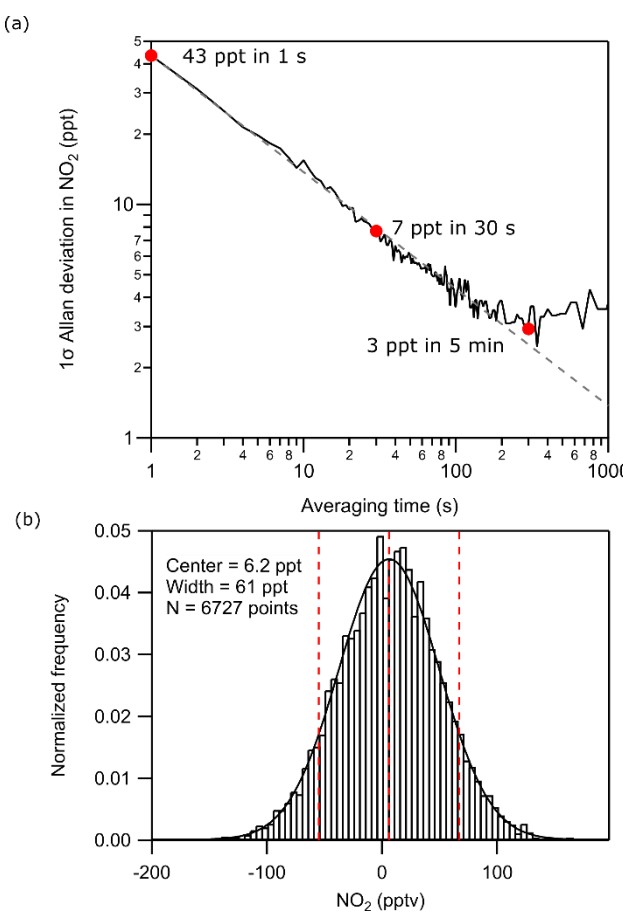






**Figure 4.** Vertical profiles of NO$_2$ during ascent (red) and descent (blue) measured during a UAV flight from 0–120 m in Boulder, Colorado. Filled circles and bars represent the average and standard deviation for measurements acquired for 10 s at each 10 m height during ascent. The averaged vertical profile of ambient temperature is shown in black.

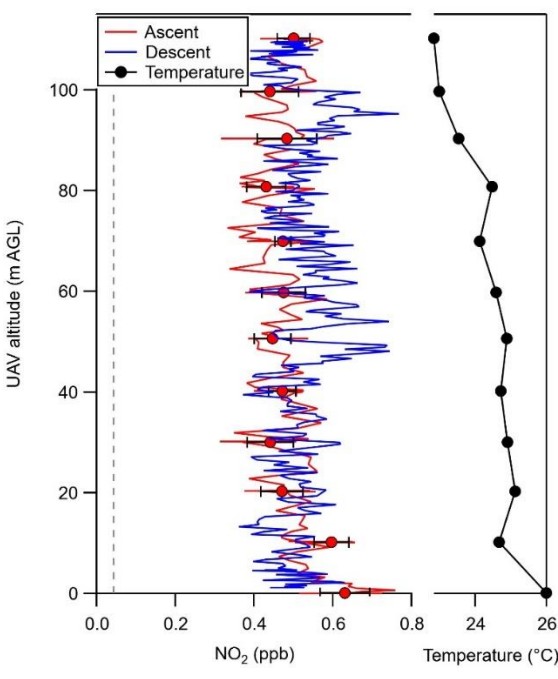
