# Peer review of "A lightweight broadband cavity-enhanced spectrometer for NO2 measurement on uncrewed aerial vehicles"

_Atmospheric Measurement Techniques, 2022_

## Referee Comment (RC1)

Journal:        Atmospheric Measurement Techniques
Submission:     amt-2022-203
DOI:            10.5194/amt-2022-203
Title:          A lightweight broadband cavity-enhanced spectrometer for $NO_2$ measurement on uncrewed aerial vehicles
Authors:        C.C. Womack et al.

The article by Womack et al. describes the development of a compact, portable, lightweight, and low power field instrument for *in situ* detection of $NO_2$, based on incoherent broadband cavity-enhanced absorption spectroscopy (IBBCEAS). The spectrometer is based on a high optical power LED at around 457 nm and a standard dispersive spectrometer of medium resolution (0.9 nm), which are used in conjunction with a high finesse 22 cm cavity ($R$=0.999963). The instrument's weight (3050 g) and power consumption (~35 W) enable its employment on an unmanned aerial vehicle (UAV), in the current case a standard commercial hexacopter with <6 kg payload.

The instrument's technical specifications are characterized in the laboratory in good detail with very few shortcomings. The instrument's applicability and initial performance on a UAV have been demonstrated successfully in this proof-of-principle study in Boulder, CO, for mixing ratios of $NO_2$ at sub-ppbv levels in vertical measurements at altitudes of up to 110 m. Based on the instrument's 1σ (laboratory-)precision of 43 pptv of $NO_2$ for an integration time of 1 s, and the stated accuracy of ~4.5%, it will be a very useful tool for airborne and other highly flexible monitoring of $NO_2$ and potentially other atmospheric species in the future.

The miniaturization of ultra-sensitive cavity enhanced absorption spectrometers (like this IBBCEAS device) enables the deployment on all sorts of highly mobile platforms, not only copter drones like in the current case. Conventional ground-based monitoring networks that typically use established standards, such a chemiluminescence detection for $NO_2$, are static and do not provide the spatial resolution required to characterize sources and/or sinks in a way that allows the advancement or validation of existing atmospheric chemistry mechanism or transport models. Innovative developments of the kind outlined by Womack et al. will clearly help to advance trace gas monitoring efforts and the investigation of the lower planetary boundary, whose spatial and temporal complexity requires the most sensitive and flexible technology to tackle current scientific questions. As far as I am aware this is the first report of an IBBCEAS instrument being employed on a drone. The merit of this innovation is that, in contrast to other alternative techniques like the more compact, lighter, and cheaper wet chemical sensors, IBBCEAS is capable of delivering reliable $NO_2$ mixing ratios at most typical atmospheric scenarios. However, subsequent campaigns will be necessary to further investigate the robustness and environmental adaptability of the new instrument - further characterization of its capabilities in the context of field monitoring is thus desirable.

In my opinion this article is basically publishable in its present form, subject to some small technical revision and some clarifications as follows:

(1) The way the spectrometer was mounted to the UAV should be explained in more detail in section 4 or in section 2.6. Only a mounting kit is mentioned. Especially the position of the inlet with respect to the rotors is of interest here. In line 104 it is merely stated that a quarter inch Teflon line extended directly above the UAV. In section 2.5 more detail on the mechanical system should be given, or put into an appendix or supplementary material.

The information in lines 236-241 cannot be put into perspective with more detail on the position of the inlet.

(2) An argument should be made in Line 104 why mirror purges are unnecessary in this case. Without mirror purges the area of applications of this compact spectrometer will be somewhat limited to not heavily polluted areas, which should be mentioned.

(3) A few more details should be given on the fitting procedure of the spectra. If this has been published earlier, then a reference should be included here. The items of interest are (i) the correlation of the fit parameters in the Levenberg-Marquardt approach, which can be an issue in comparison to other analysis methods, (ii) the dependence of the results and their error on the fit range, and (iii) the choice and justification of the wavelength-dependent weighting factors mentioned in the text. How were these factors chosen, what criteria were applied?

(4) The error estimation in section 5.1 is not very conservative and has gaps. Nonetheless the overall accuracy is stated as 4.5% in the summary and Table 1. I recommend to be a bit more cautious here. I have my doubts that the instrument will really live up to this accuracy in field campaigns.

**Minor suggestions and comments:**
Line 15: … has a power consumption of less …
Line 83: The authors state a temperature of 22.5±0.05 °C. The number of significant figures is inconsistent here. How was this temperature accuracy established?
Line 84: Acronym TEC = ThermoElectric Cooler?
Line 131/132: It would be good to also state the information given in form of a duty cycle time.
Line 147: "…38 min minus (3.6 min kg$^{-1}$ × payload mass)" may be nicer expressed in form of a proper equation.
Line 169: What is a typical time for the zero air calibration measurements?
Line 180: The number of publication of the $NO_2$ cross-section is large. What is the reason for using the values by Vandaele et al. (1998) in this case?
Line 224: Where is the optical extinction shown in Figure 3?
Line 230: "…0-120 m…" Figure 4 only shows 0-110 m. If data are available for only up to 110 m altitude, then this should be corrected; also in the abstract and throughout the text where 120 m are stated.
Line 236: "… have been completed by McKinney et al. 2019". Include the reference here directly in the text.
Line 246: "The mACES inlet was located vertically above the UAV, in the region that is modeled to have approximately laminar flow." This should be shown – see also item (1) above. More detail required here.
Line 251: "The measurement precision of 43 ppt $NO_2$ in 1 s suggests that the observed variability within the vertical profile represents real $NO_2$ variation,…". I agree that the precision of 43 pptv would suggests that, however, it is not quite clear how meaningful this measurement is. On the descent (blue trace in Fig. 4) the data points represent averages over a vertical distance of 0.5 m. On the ascent (red trace) it is not quite clear how the measurement were performed in between altitude steps of 10 m (see also paragraph starting Line 196 ff.). Please discuss in some more detail
Comment: The hovering at 10 m intervals and estimating the error as given by the error bars in Figure 4 is meaningful appear the most meaningful.

Tables 1: precision $\rightarrow$ Precision (1 sigma)

Figure 1: (b) - Acronym MFM is not defined. (c) & (d) - The size scale of the instrument cannot be made out appropriately.

Figure 3 caption: "…of fitted $NO_2$…  . … of fitted $NO_2$" is too casual. Please rephrase. Throughout the manuscript (incl. Figures) mixing ratios should be stated parts per number by volume: ppm $\rightarrow$ ppmv, ppb $\rightarrow$ ppbv etc.

Figure 4: The measurement conditions on the descent (continuous at constant speed) should also be stated in the figure caption.

---

## Author Comment (AC1)

**Referee #1:**

The article by Womack et al. describes the development of a compact, portable, lightweight, and low power field instrument for *in situ* detection of NO2, based on incoherent broadband cavity-enhanced absorption spectroscopy (IBBCEAS). The spectrometer is based on a high optical power LED at around 457 nm and a standard dispersive spectrometer of medium resolution (0.9 nm), which are used in conjunction with a high finesse 22 cm cavity (*R*=0.999963). The instrument's weight (3050 g) and power consumption (~35 W) enable its employment on an unmanned aerial vehicle (UAV), in the current case a standard commercial hexacopter with <6 kg payload.

The instrument's technical specifications are characterized in the laboratory in good detail with very few shortcomings. The instrument's applicability and initial performance on a UAV have been demonstrated successfully in this proof-of-principle study in Boulder, CO, for mixing ratios of NO2 at sub-ppbv levels in vertical measurements at altitudes of up to 110 m. Based on the instrument's  $1\sigma$  (laboratory-)precision of 43 pptv of NO2 for an integration time of 1 s, and the stated accuracy of ~4.5%, it will be a very useful tool for airborne and other highly flexible monitoring of NO2 and potentially other atmospheric species in the future.

The miniaturization of ultra-sensitive cavity enhanced absorption spectrometers (like this IBBCEAS device) enables the deployment on all sorts of highly mobile platforms, not only copter drones like in the current case. Conventional ground-based monitoring networks that typically use established standards, such a chemiluminescence detection for NO2, are static and do not provide the spatial resolution required to characterize sources and/or sinks in a way that allows the advancement or validation of existing atmospheric chemistry mechanism or transport models. Innovative developments of the kind outlined by Womack et al. will clearly help to advance trace gas monitoring efforts and the investigation of the lower planetary boundary, whose spatial and temporal complexity requires the most sensitive and flexible technology to tackle current scientific questions. As far as I am aware this is the first report of an IBBCEAS instrument being employed on a drone. The merit of this innovation is that, in contrast to other alternative techniques like the more compact, lighter, and cheaper wet chemical sensors, IBBCEAS is capable of delivering reliable NO2 mixing ratios at most typical atmospheric scenarios. However, subsequent campaigns will be necessary to further investigate the robustness and environmental adaptability of the new instrument - further characterization of its capabilities in the context of field monitoring is thus desirable.

In my opinion this article is basically publishable in its present form, subject to some small technical revision and some clarifications as follows:

We thank the reviewer for these comments on our manuscript. We have addressed the comments listed below, and provide changes listed in blue. We also provide a tracked-changes version of the manuscript. Line numbers in our response refer to this manuscript version.

(1) The way the spectrometer was mounted to the UAV should be explained in more detail in section 4 or in section 2.6. Only a mounting kit is mentioned. Especially the position of the inlet with respect to the rotors is of interest here. In line 104 it is merely stated that a quarter inch Teflon line extended directly above the UAV. In section 2.5 more detail on the mechanical system should be given, or put into an appendix or supplementary material. The information in lines 236-241 cannot be put into perspective with more detail on the position of the inlet.

We now realize that the "expansion mounting kit" mentioned in the manuscript is actually a standard frame on the underside of the drone, and so we have changed Line 199 to read: "... to the Matrice 600 Pro underside carbon-fiber rectangular mounting frame". Additionally, we updated Line 200 to clarify how the instrument is secured to the drone: "...using the four quick release brackets, shown in yellow in Figure 1c." Additionally, the Figure 1 caption now includes the text: "(c) Model of the instrument, with quick-release clamps for mounting the instrument to the drone highlighted in yellow;"

We opted for a simple inlet in this instrument paper, consisting of just a Teflon tube protruding above the propellers, but still within the estimated propeller wash. For some sampling applications, we anticipate this will be sufficient, while for others, a more complicated sampling setup may be desired. But we decided that a detailed investigation of sampling from drones would be beyond the scope of this paper, and would concern all types of drone-based instrumentation, not just our NO2 instrument. Therefore, we opted to focus on the mACES instrument, and keep the inlet simple. We have added further clarification about it on lines 105 - 107: "During the test flights described here, the sampling inlet was a 0.635 cm OD Teflon tube that extended 0.2 m directly above the UAV rotors and was secured to the drone's antenna.", and on lines 118 - 119: "The residence time in the sampling inlet line is estimated as 0.2 s and therefore did not add significantly to the total residence time." Additionally, we added the following at line 257 - 258 to discuss possible upgrades to the inlet, if sampling outside the prop wash is desired. "If sampling outside the propeller wash is desired, a lightweight sideways sampling inlet arm could be added to the payload."

(2) An argument should be made in Line 104 why mirror purges are unnecessary in this case. Without mirror purges the area of applications of this compact spectrometer will be somewhat limited to not heavily polluted areas, which should be mentioned.

It is true that mirror purges are usually very helpful in keeping mirrors clean, but they are difficult to put on a drone-based instrument due to the need for a clean zero air source, and the

instrument can be run without them in most situations for shorter periods of time. It has been our experience that sudden changes in mirror cleanliness occur only when large particles or specks of dust in unfiltered air impact on the mirrors. This is prevented in mACES with the upstream Teflon filters. Loss of cleanliness due to a high concentration of semi-volatile gases coating the mirrors can still occur, but tends to happen gradually, if at all. Such changes are typically much slower than the frequency of reflectivity measurements. We note that commercial cavity enhanced instruments (e.g. Picarro, LGR) typically do not use mirror purges and are stable over long periods of time. Therefore, accurate measurements can still occur in heavily polluted areas, but could potentially require more frequent mirror cleanings, if the measured mirror reflectivity begins to drop. We have changed the text to better explain this on lines 109 - 113: "Following the approach described in Min et al. (2016), mirror purges for the cavity mirrors are not used, as they require a bulky and relatively heavy zero air source. The lack of mirror purges makes maintaining mirror cleanliness, monitored using the measured mirror reflectivity (see Sect. 3), especially critical. Mirrors are removed and cleaned as necessary, approximately monthly, though may require more frequent cleanings when sampling in polluted atmospheres."

(3) A few more details should be given on the fitting procedure of the spectra. If this has been published earlier, then a reference should be included here. The items of interest are (i) the correlation of the fit parameters in the Levenberg-Marquardt approach, which can be an issue in comparison to other analysis methods, (ii) the dependence of the results and their error on the fit range, and (iii) the choice and justification of the wavelength-dependent weighting factors mentioned in the text. How were these factors chosen, what criteria were applied?

The Levenberg-Marquardt algorithm (LMA) has been well-established for both CES and DOAS techniques. To give readers more context, we have now included a reference to Kraus 2006, and Platt 2009 at line 193.

- (i) Because the LMA has been so widely used, we felt an in-depth investigation of the correlation of fit parameters with the LMA technique, versus something like singular value decomposition, is therefore beyond the scope of this paper.
- (ii) We selected the maximum fit window for which the optical cavity has sufficiently high precision. Our primary metric is whether the residual of the fit is free of structural features, which indicates that the fitted parameters are consistent throughout the entire window. While we have not explicitly characterized the dependence of the fitting results on the fit window, a future manuscript is planned that will address this in some detail for the CES technique in general.
- (iii) The weighting factor is the wavelength-dependent error variance of the extinction measurement,  $\sigma^2(\lambda)$ . Therefore, it adds additional weight to the fit at wavelengths where the spectrum is most accurate.

To address all three comments, we have updated the paragraph starting at line 192 to read: "The spectral fitting to determine  $N_i$  and  $p(\lambda)$  in Eqn. 3 used custom software developed in Igor Pro (WaveMetrics Inc., Portland, OR, USA) and based on Levenburg-Marquardt least-squares linear fitting (Kraus, 2006; Platt et al., 2009). The fit was optimized between 430 and 476.5 nm, the maximum fitting window that also minimizes spectral features in the residual spectrum. The algorithm also used the measurement error variance as a wavelength-dependent weighting factor to prioritize the fit in the spectral region where the instrument performance was most precise."

(4) The error estimation in section 5.1 is not very conservative and has gaps. Nonetheless the overall accuracy is stated as 4.5% in the summary and Table 1. I recommend to be a bit more cautious here. I have my doubts that the instrument will really live up to this accuracy in field campaigns.

We feel confident in how this instrument has been ruggedized against vibrations and temperature swings, so we don't expect the precision and accuracy to be too much different than what was measured in the lab. However, the reviewer is correct that it is always a possibility, so we have added the following to line 229: "Field measurements may show more variability than laboratory measurements, but this variability can be monitored during the regular zero air additions." We will continue to monitor the precision of the instrument during the regular calibrations, and will plan to provide a dynamic estimate of the instrument precision whenever reporting data.

**Minor suggestions and comments:**

Line 15: ... has a power consumption of less ... *This has been fixed.*

Line 83: The authors state a temperature of  $22.5\pm0.05$  °C. The number of significant figures is inconsistent here. How was this temperature accuracy established? For the LED, the temperature accuracy is not especially important, but the stability is. The 0.05 °C figure comes from the observed variability in the measured LED temperature, which nearly always stayed between 22.45 and 22.55 °C. We have changed the text to better represent the significant figures: "temperature-controlled at  $22.50\pm0.05$  °C"

Line 84: Acronym TEC = ThermoElectric Cooler? *Yes, that is correct, and is stated in parentheses on line 85.*

Line 131/132: It would be good to also state the information given in form of a duty cycle time. We opted to put the statement about the 99% duty cycle at line 101: "However, the QE Pro weighs only 1.15 kg compared to 6.8 kg, requires no physical shutter, and can read out the vertically-integrated CCD in 0.002 s allowing a spectrometer duty cycle of 99% for a 0.15 s integration time."

Line 147: "...38 min minus (3.6 min kg-1  $\times$  payload mass)" may be nicer expressed in form of a proper equation.

This has been changed in the text and labeled as Equation 1. All subsequent equations and references to these equations have been incremented.

Line 169: What is a typical time for the zero air calibration measurements? We overflow zero air and helium for approximately 15 seconds each, as described in Section 4 (Line 202)

Line 180: The number of publication of the NO2 cross-section is large. What is the reason for using the values by Vandaele et al. (1998) in this case?

We used the Vandaele (1998) cross section because it is the recommended value by the 2010 JPL evaluation (https://jpldataeval.jpl.nasa.gov/)

Line 224: Where is the optical extinction shown in Figure 3?

This was a typographical error from a previous draft. That line should have referred to the Allan deviation of just the retrieved NO2 concentration. We thank the reviewer for catching this. The line now reads "Allan deviation plots (Werle et al., 1993) were calculated for the retrieved NO2 concentrations, to quantify the precision and drift as a function of time. Fig. 3a and 3b show the Allan deviation and normalized histogram for the retrieved NO2 concentrations during the zero air measurements."

Line 230: "...0-120 m..." Figure 4 only shows 0-110 m. If data are available for only up to 110 m altitude, then this should be corrected; also in the abstract and throughout the text where 120 m are stated.

Thank you for pointing this out. Yes, the maximum FAA-allowable height is 120 meters AGL, but to stay within that limit, we only flew the drone to 110 m as an additional safety precaution. We have changed the text from 120 to 110 m in several locations (Lines 65, 206, 240, 260, 273, and the Figure 4 caption) to reflect the true bounds of the measured profile.

Line 236: "... have been completed by McKinney et al. 2019". Include the reference here directly in the text.

This has been fixed.

Line 246: "The mACES inlet was located vertically above the UAV, in the region that is modeled to have approximately laminar flow." This should be shown – see also item (1) above. More detail required here.

Please see our response to item (1).

Line 251: "The measurement precision of 43 ppt NO2 in 1 s suggests that the observed variability within the vertical profile represents real NO2 variation,…". I agree that the precision of 43 pptv would suggests that, however, it is not quite clear how meaningful this measurement is. On the descent (blue trace in Fig. 4) the data points represent averages over a vertical distance of 0.5 m. On the ascent (red trace) it is not quite clear how the measurement were performed in between altitude steps of 10 m (see also paragraph starting Line 196 ff.). Please discuss in some more detail

Between the 10 m intervals, the drone ascended at a rate of 1 m s-1. But in addition to different ascent/descent rates, the ascent and descent profiles are measuring slightly different air masses, since they are displaced in time. The NO2 variability we referred to was in reference to these changes in the true NO2 concentration at different times. To clarify this, we have added the following to line 263: "which is expected since the measurement site is 440 m from a busy road, and the ascent and descent profiles are displaced several minutes in time". Additionally, we added the text "The ascent rate between intervals was 1.0 m s-1" to line 207, and the text "with a 1 m s-1 ascent rate between each level leg" to the Figure 4 caption.

Comment: The hovering at 10 m intervals and estimating the error as given by the error bars in Figure 4 is meaningful appear the most meaningful.

We reported error based on laboratory tests, rather than the measured vertical profile, to avoid any additional contribution from real NO2 variability, which is not true instrument error.

Tables 1: precision  $\rightarrow$  Precision (1 sigma) *This has been fixed.*

Figure 1: (b) - Acronym MFM is not defined. (c) & (d) - The size scale of the instrument cannot be made out appropriately.

We have now defined MFM (mass flow meter) in the caption. Part (b) of the caption now reads "Block diagram of the flow system, showing the ground calibration unit, inlet filter, sample cell, mass flow meter (MFM), pressure sensor, and pump;"

To clarify the size scale, we have added the following text to part (d) of the caption: "The total dimensions of the instrument are approximately  $45 \times 20 \times 20$  cm (length x width x height)."

Figure 3 caption: "...of fitted NO2... of fitted NO2" is too casual. Please rephrase. Throughout the manuscript (incl. Figures) mixing ratios should be stated parts per number by volume: ppm  $\rightarrow$  ppmv, ppb  $\rightarrow$  ppbv etc.

We have changed the language from "the fitted NO2" to "the retrieved NO2 concentration" in both locations.

As for ppb vs ppbv, we recognize that this has been a much-debated topic. Our reasoning for using ppb instead of ppbv is that ppbv is only strictly meaningful for ideal gases, for which the volume of mixed gases is conserved. However, we also want to emphasize that the unit is a mole fraction, rather than a mass fraction. To remove any ambiguity, we have changed the text to define ppb as a unit of mole mixing ratio at line 185: "(27.2 parts per million by mole (ppm), diluted in zero air to a mole mixing ratio of approximately 100 parts per billion by mole (ppb)". We hope this clarifies how we define ppb, ppm, and ppt here.

Figure 4: The measurement conditions on the descent (continuous at constant speed) should also be stated in the figure caption.

The following sentence was added to the Figure 4 caption: "During descent, the drone descended at a constant rate of  $0.5 \text{ m s}^{-1}$ ."

---

## Author Comment (AC2)

**Referee #2:**

The manuscript, "A lightweight broadband cavity-enhanced spectrometer for NO2 measurement on uncrewed aerial vehicles," by Womack et al. describes the design and performance of a new, compact instrument for atmospheric measurements of NO2. NO2 plays multiple roles in the atmosphere, including as a pollutant and key player in oxidation chemistry in the troposphere. The availability of inexpensive uncrewed aerial vehicles (UAVs) has increased the need for smaller, lighter, and less expensive instruments for measuring atmospheric trace gases. Previously, NO2 has been measured with large, expensive optical instruments. Small electrochemical sensors exist but lack sufficient sensitivity for atmospheric measurements. Hence this work is of significant importance to the atmospheric chemistry community. The instrument design utilizes broadband cavity enhanced absorption spectroscopy, and is similar to an existing larger instrument by the same team with a demonstrated track record of NO2 measurements. The paper is well written, and the description of the instrument is detailed and clear. The method has been tested successfully in preliminary flights, the results of which are included. I have only a few small comments and I recommend publication following minor revisions.

*We thank the reviewer for these comments on our manuscript. We have addressed the comments listed below, and provide changes listed in* blue. *We also provide a tracked-changes version of the manuscript. Line numbers in our response refer to this manuscript version.*

Equation 1: In equation 1, there is no Δ in front of α ray,ZA in the first parentheses, but in the description in the text, there is a Δ. This seems inconsistent and should be fixed. (In the equation in the referenced paper by Min et al., (2016), there is no Δ in front of α ray,ZA or α ray,sample. The Δ appears in Δ α ray, which seems correct as it is the difference between the two.)
*We thank the reviewer for catching this. The Δ in the text is indeed a typo and has been removed.*

Line 192: Reference is made to operating the instrument with the LED off, but no description of the instrument control is given. How is it controlled during flight, or on the ground? Is there an operation algorithm, or does everything turn on when powered?

*Currently, the LED and pump are both powered on and off by physically disconnecting their power cables from the main power source. As the calibrations are completed on the ground, electronic/remote control wasn't necessary, but future efforts will focus on developing this capacity for added ease of use. We would prefer to keep this description simple, but for clarity, we have changed line 200 to read* "We then power on the instrument and record dark background spectra with no LED light."

Section 5.2: The instrument accuracy based on standards is very good, but it would be interesting to include an intercomparison of this instrument with the standard, aircraft-based BBCES instrument as well to further test the accuracy.

*Future field studies with both instruments can be used to compare them directly while sampling ambient air at a range of conditions. For now, we note the fact that both the original aircraft-based ACES and mACES use an identical detection technique and analysis approach, and we have confirmed their responses to standard NO2 concentrations (in this work, as in Min et al. 2016). We have included the following paragraph on line 287:* "Despite its small size, the accuracy of this method is comparable to that of other spectroscopically based instruments (e.g., CRDS (Wild et al., 2014), LIF (Thornton et al., 2000)) and of research-grade photolytic conversion of $NO_2$ followed by detection of NO (Pollack et al., 2010). The versatility of mACES may facilitate intercomparisons of research grade and monitoring network $NO_2$ and $NO_x$ instruments."

Line 225: The text says that Figure 3 shows the Allan deviation for the optical extinction and retrieved NO2, but the figure only seems to show the retrieved NO2.

*This was a typographical error from a previous draft. That line should have referred to the Allan deviation of just the retrieved $NO_2$ concentration. We thank the reviewer for catching this. The line now reads* "Allan deviation plots (Werle et al., 1993) were calculated for the retrieved $NO_2$ concentrations, to quantify the precision and drift as a function of time. Fig. 3a and 3b show the Allan deviation and normalized histogram for the retrieved $NO_2$ concentrations during the zero air measurements."

Line 245: Little description is given on the inlet used, just that it extends above the UAV. More information is needed. How far above? What is its configuration? What is the residence time of the sample prior to entering the detection volume? This is of particular importance because of the disturbance to the surrounding air by the UAV, which in turn affects how the measurements can be interpreted.

*(Response to Referee #1, item (1) is reproduced here): We opted for a simple inlet in this instrument paper, consisting of just a Teflon tube protruding above the propellers, but still within the estimated propeller wash. For some sampling applications, we anticipate this will be sufficient, while for others, a more complicated sampling setup may be desired. But detailed investigation of sampling from drones is beyond the scope of this paper, and concerns all types of drone-based instrumentation, not just our $NO_2$ instrument. Therefore, we opted to focus on the mACES instrument, and keep the inlet simple. We have added further clarification about it on lines 105 - 107:* "During the test flights described here, the sampling inlet was a 0.635 cm OD Teflon tube that extended 0.2 m directly above the UAV rotors and was secured to the drone's antenna.", *and on lines 118 - 119:* "The residence time in the sampling inlet line is estimated as

*0.2 s and therefore did not add significantly to the total residence time."* Additionally, we added the following at line 257 - 258 to discuss possible upgrades to the inlet, if sampling outside the prop wash is desired. *"If sampling outside the propeller wash is desired, a lightweight sideways sampling inlet arm could be added to the payload."*